# Spike-RetinexFormer: Rethinking Low-light Image Enhancement with Spiking Neural Networks

**Hongzhi Wang**
School of Software Technology
Zhejiang University
Ningbo, China
whzzju@zju.edu.cn

**Xiubo Liang**[*]
School of Software Technology
Zhejiang University
Ningbo, China
xiubo@zju.edu.cn

**Jinxing Han**
School of Software Technology
Zhejiang University
Ningbo, China
hanjx@zju.edu.cn

**Weidong Geng**
College of Computer Science and Technology
Zhejiang University
Hangzhou, China
gengwd@zju.edu.cn

## Abstract

Low-light image enhancement (LLIE) aims to improve the visibility and quality of images captured under poor illumination. However, existing deep enhancement methods often underemphasize computational efficiency, leading to high energy and memory costs. We propose **Spike-RetinexFormer**, a novel LLIE architecture that synergistically integrates Retinex theory, spiking neural networks (SNNs) and a Transformer-based design. Leveraging sparse spike-driven computation, the model reduces theoretical compute energy and memory traffic relative to ANN counterparts. Across standard benchmarks, the method matches or surpasses strong ANNs (25.50 dB on LOL-v1; 30.37 dB on SDSD-out) with comparable parameters and lower theoretical energy. Our work pioneers the synergistic integration of SNNs into Transformer architectures for LLIE, establishing a compelling pathway toward powerful, energy-efficient low-level vision on resource-constrained platforms.

## 1 Introduction

Capturing images in low-light conditions is challenging due to limited photons, sensor noise, and constrained dynamic range. These factors produce dark, noisy images that hinder downstream tasks like detection or recognition. LLIE techniques seek to brighten such images and restore details, enabling better visual quality and utility. Early approaches relied on heuristic image processing (gamma correction, histogram equalization) and the Retinex theory of human vision [1], which decomposes an image into reflectance and illumination. Traditional Retinex-based algorithms [2, 3] improved visibility by estimating a smooth illumination map and enhancing the reflectance, but often suffered from artifacts (haloing, color distortion) and were limited by manual parameter tuning.

The advent of deep learning spurred significant advances in LLIE. Convolutional neural networks (CNNs) have been trained to directly map dark images to brighter ones, outperforming classical methods in handling noise and complex artifacts. Representative works include LLNet [4], one of the first deep autoencoders for natural low-light enhancement, and LightenNet [5] for weak illumination images. More specialized models integrated Retinex theory, such as RetinexNet [6] which learned to decompose an image into reflectance and illumination and enhance them with

---

[*]Corresponding author

39th Conference on Neural Information Processing Systems (NeurIPS 2025).

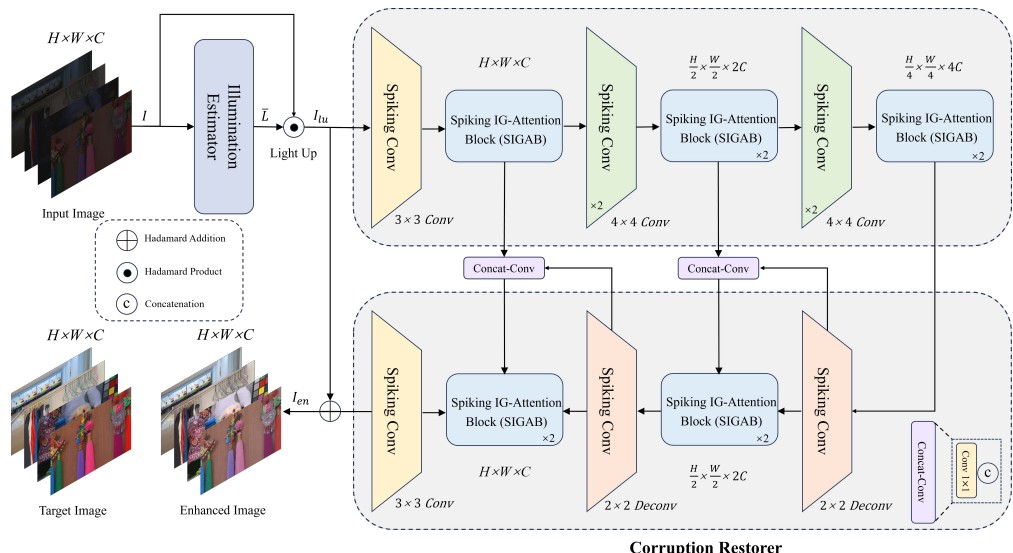

Figure 1: Architecture of the Spike-RetinexFormer. It consists of two main stages: (1) the Spiking Illumination Estimator, which predicts illumination features and a light-up map to adjust initial brightness, (2) the Spiking Corruption Restorer, which utilizes a multi-scale U-Net structure with Spiking Illumination-Guided Attention.

deep networks. Many subsequent methods built on this idea, proposing improved Retinex-based networks with attention or refinement modules [7, 8, 9]. Unsupervised and zero-reference techniques have also emerged: Zero-DCE [10, 11] optimizes a deep curve mapping without ground truth, while EnlightenGAN [12] uses generative adversarial learning to enhance lighting without paired data. Recent trends include normalizing flow models [13] and diffusion models [14] for LLIE, and Transformer-based architectures [15] to capture global illumination context. Despite improved enhancement quality, most deep LLIE methods are computationally heavy and energy-demanding[16, 17], which limits deployment on resource-constrained devices.

Meanwhile, SNNs have gained interest as the third generation of neural networks, offering event-driven computation inspired by biological neurons [18]. Neurons communicate via sparse binary spikes, leading to significantly reduced power consumption on neuromorphic hardware [19]. SNNs have achieved competitive performance on classification [20] and object detection [21] tasks with far lower energy usage than standard ANN models. However, relatively few works have applied SNNs to low-level vision: LLIE is a continuous-tone regression problem (enhancing pixel intensities), whereas most prior SNN works focused on classification or recognition with discrete labels. Directly applying SNNs to image enhancement must handle fine-grained color and illumination adjustments, requiring high precision despite the coarse spike-based computations. [22] encoded pixel intensities into spike latencies and used a recurrent SNN (with ConvLSTM) for unsupervised LLIE. This demonstrates that SNNs can gradually capture image structure via spike timing.

Transformers [23] have revolutionized many vision tasks due to their self-attention mechanism capturing long-range dependencies. Vision Transformers [24] and related models have shown excellent performance in recognition and even low-level image restoration [25, 26]. For LLIE, Transformers can globally model illumination variation and object context, yielding more balanced enhancement [15]. Merging the energy efficiency of SNNs with the representational strength of Transformers is a promising direction that remains underexplored. Recent studies have started integrating spiking neurons into Transformer models [27, 28, 29, 30, 31], mainly for classification or sequential data processing. These spiking Transformers achieve comparable accuracy to standard Transformers while greatly reducing Multiply-Accumulate operations via spike-based attention. This progress motivates our approach to design a spiking Transformer for LLIE.

In this paper, we propose Spike-RetinexFormer, a SNN architecture for LLIE. Spike-RetinexFormer is a spike-driven variant of RetinexFormer that re-instantiates the one-stage Retinex parameterization with temporal spike coding, Leaky Integrate-and-Fire (LIF [32]) neurons, surrogate-gradient training,

and an event-driven attention mechanism. To our knowledge, this is among the first feedforward spiking Transformer-style network tailored to LLIE. By coupling Retinex-based illumination modeling with event-driven computation, the approach aims at high-fidelity enhancement under constrained compute and energy budgets. Specifically, a spiking illumination estimator predicts a light-up map and illumination features consistent with the Retinex reparameterization, while a Retinex restoration module employs spiking illumination-guided attention (SIGA) to aggregate long-range context and suppress noise. Our network processes images over $T$ time steps, accumulating an enhanced output from spiking neuron responses. In summary, our contributions to the community include:

- We introduce Spike-RetinexFormer, which instantiates a one-stage Retinex enhancement pipeline entirely using spiking primitives and is trained with surrogate gradients, demonstrating that continuous image restoration can be effectively addressed in the spike domain.
- We develop a spiking illumination-guided attention mechanism—implemented via spike-coincidence affinities, illumination-gated sparsification, and binary value routing—and integrate it into the Retinex framework to enable long-range interactions without forming dense $QK^\top$ matrices or softmax normalization.
- Across standard LLIE benchmarks, Spike-RetinexFormer attains competitive enhancement quality relative to representative ANN counterparts while using comparable parameters and fewer theoretical FLOPs.

## 2 Related Works

### 2.1 Low-Light Image Enhancement

Early methodologies for low-light image enhancement drew upon traditional image processing techniques and models of human visual perception, prominently featuring the Retinex theory [1], which models an image as a product of reflectance and illumination components. Subsequent developments, such as Multi-Scale Retinex with Color Restoration (MSRCR) [2] and LIME [3], focused on estimating illumination maps to improve image visibility; however, these approaches necessitated manual parameter tuning and were susceptible to artifact generation. The emergence of deep learning marked a paradigm shift, introducing data-driven approaches that often yielded superior performance. CNN-based models, exemplified by LLNet [4] and LightenNet [5], utilized large-scale datasets (LOL [6]) to learn the image enhancement mapping directly from data. While not exclusively focused on LLIE, Ignatov et al. [33] explored broader applications of CNNs in image enhancement, contributing to the foundational understanding in the field. The principles of Retinex theory were subsequently incorporated into deep learning frameworks. Notable models include RetinexNet [6] and its variants [7, 8], which explicitly perform image decomposition into illumination and reflectance, with applications extending to specialized domains such as underwater imaging [9] and back-lit scene enhancement.

To address the limitations associated with paired training data, unsupervised and semi-supervised methods were developed. For instance, Zero-DCE [10] and its subsequent refinement [11] utilized non-reference loss functions, while EnlightenGAN [12] and DeepExposure [34] employed generative models for unpaired learning scenarios. Yang et al. [35] proposed a semi-supervised framework designed to strike a balance between image fidelity and perceptual quality. Recent investigations have focused on leveraging sophisticated neural architectures. Models based on normalizing flows [13] and diffusion probabilistic models [14] have demonstrated capabilities for high-fidelity image enhancement, albeit often at a significant computational cost. Inspired by their success in other computer vision tasks such as super-resolution [25, 26], Transformer architectures and attention mechanisms have also been adapted for LLIE, as evidenced by works like [15]. In a related vein, Tang et al. [36] introduced a method focusing on disentangling various image components to achieve more flexible and controllable enhancement.

### 2.2 Spiking Neural Networks

SNNs are a class of biologically-inspired computational models that process information through discrete temporal events, termed spikes, contrasting with the continuous activation values characteristic of conventional Artificial Neural Networks (ANNs)[37]. Within SNNs, individual neurons integrate input currents over time, generating an output spike when their internal membrane potential surpasses

a predefined threshold. This event-driven operational paradigm offers the potential for substantial energy efficiency, as computationally expensive multiply-accumulate operations prevalent in ANNs are often replaced by simpler arithmetic operations (additions), and power consumption is predominantly associated with active spiking events. Maass [18] conceptualized SNNs as the third generation of neural network models, and subsequent research has established their viability as an alternative to ANNs in specific application domains. For instance, [19] demonstrated that neuromorphic hardware platforms executing SNNs can achieve energy efficiencies orders of magnitude greater than those of Graphics Processing Units (GPUs) processing functionally equivalent ANN models.

The practical training of SNNs was initially impeded by the non-differentiable nature of the spike generation mechanism. However, methodologies such as ANN-to-SNN conversion and surrogate gradient learning have facilitated the effective training of deep SNN architectures [20].Consequently, SNNs have demonstrated competitive performance on established image classification benchmarks [38, 20, 39, 40, 41, 42] and have been successfully applied to more complex vision tasks, including object detection (Spiking-YOLO [21]). Notwithstanding these advancements, the application of SNNs to low-level vision tasks requiring continuous-valued outputs, such as image enhancement, remains relatively underexplored. A principal challenge in this context is the generation of high-resolution, analog-like outputs (an enhanced image) from discrete spike trains. This often necessitates sophisticated encoding schemes for pixel intensities, such as rate or temporal coding, thereby introducing additional representational and computational complexity. Addressing this, [22] investigated LLIE by encoding pixel intensities into spike latencies, which were then processed by a recurrent SNN. Their findings demonstrated the feasibility of image enhancement using this paradigm, achieving notable computational reductions compared to some ANN counterparts. These results provide preliminary evidence for the capability of SNNs in image enhancement tasks.

## 3 Method

### 3.1 Spiking Retinex-Based Enhancement Framework

Following Retinex theory, a low-light RGB image $I \in [0, 1]^{H \times W \times 3}$ is modeled as the Hadamard product of a reflectance image $R \in [0, 1]^{H \times W \times 3}$ and an illumination map $L \in [0, 1]^{H \times W}$:

$$I = R \odot L. \tag{1}$$

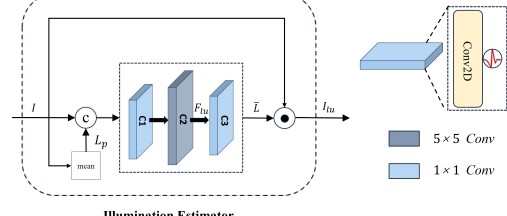

Directly obtaining a lit image by element-wise division is numerically fragile when $L$ is small. Instead, we introduce a light-up map $\bar{L}$ that approximates $L^{-1}$ and enforces the constraint $\bar{L} \odot L \simeq 1$. Multiplying by $\bar{L}$ yields a lit-up image

Figure 2: Spiking illumination estimator $\mathcal{E}$.

$$I_{\mathrm{lu}} = I \odot \bar{L} = R + C, \tag{2}$$

where $C \in \mathbb{R}^{H \times W \times 3}$ collects the overall corruption introduced by sensor noise, quantization, color shifts, and the light-up process itself. We realize (2) in one stage with a spiking illumination estimator and a spiking corruption restorer. We adopt a one-stage Retinex-based framework with two spike-driven modules:

$$(I_{\mathrm{lu}}, F_{\mathrm{lu}}) = E(I, L_p), \qquad I_{\mathrm{en}} = G(I_{\mathrm{lu}}, F_{\mathrm{lu}}), \tag{3}$$

where $E$ denotes the *spiking illumination estimator* and $G$ the *spiking corruption restorer*. The illumination prior $L_p \in \mathbb{R}^{H \times W \times 1}$ is the channel-average mean of the input image:

$$L_p = \mathrm{mean}(I), \tag{4}$$

and both $E$ and $G$ are implemented with LIF spiking neurons and operate over $T$ discrete time steps; gradients are estimated via surrogate functions.

As illustrated in Fig. 2, $E$ is a shallow spike-driven CNN that maps $[I \parallel L_p] \in \mathbb{R}^{H \times W \times 4}$ to a light-up image and illumination features:

C1: $1 \times 1$ fusion $\rightarrow$ C2: depth-wise $5 \times 5$ (light-up features) $\rightarrow$ C3: $1 \times 1$ projection.

$G$ is a U-Net style encoder–decoder with skip connections, implemented entirely with spiking layers. At each decoder stage, we insert a SIGA block that consumes the current feature maps together with the scale-aligned $F_{\mathrm{lu}}$ (from (3)), injecting long-range, lighting-aware context without dense softmax attention. A lightweight spike-driven head predicts a residual $\Delta I$ so that

$$I_{\mathrm{en}} = I_{\mathrm{lu}} + \Delta I. \tag{5}$$

The whole pipeline (3) is trained end-to-end with surrogate gradients.

### 3.2 Spiking Neuron Model and Training

We use LIF neurons with soft reset and a single, shared time budget of $T$ steps for both training and inference. For each layer $\ell$ and time step $t \in \{1, \ldots, T\}$,

$$U^{(\ell)}[t] = \lambda^{(\ell)} H^{(\ell)}[t-1] + I^{(\ell)}[t], \quad S^{(\ell)}[t] = \Theta\big(U^{(\ell)}[t] - V_{\mathrm{th}}^{(\ell)}\big), \quad H^{(\ell)}[t] = U^{(\ell)}[t] - V_{\mathrm{th}}^{(\ell)} S^{(\ell)}[t], \tag{6}$$

with $H^{(\ell)}[0] = 0$, leak $\lambda^{(\ell)} \in (0,1)$, and a fixed threshold $V_{\mathrm{th}}^{(\ell)}$.[2]

**Input encoding and $T$-normalization.**

The illumination estimator $E$ (see (3)) is executed first and receives a time-shared, $T$-normalized current

$$I_E^{(0)}[t] = \frac{1}{T} \Phi_{\mathrm{in}}\big(I, L_p\big), \qquad \Phi_{\mathrm{in}}(X, Y) = \kappa\, \phi(X) \,\|\, \kappa_p\, \phi(Y), \quad \phi(z) = \log\big(1 + \zeta z\big), \tag{7}$$

where $\|$ denotes channel concatenation. After $E$ predicts the light-up map $\bar{L}$ and features $F_{\mathrm{lu}}$ (cf. (3)), we form $I_{\mathrm{lu}} = I \odot \bar{L}$ (cf. (2)) and its prior $L_p^{\mathrm{lu}} = \mathrm{mean}(I_{\mathrm{lu}})$. The decoder $G$ then uses

$$I_G^{(0)}[t] = \frac{1}{T} \Phi_{\mathrm{in}}\big(I_{\mathrm{lu}}, L_p^{\mathrm{lu}}\big), \tag{8}$$

which keeps the total injected charge approximately invariant to $T$. We use $\zeta = 6$ and $(\kappa, \kappa_p) = (1.0, 0.5)$ in all experiments.

**Illumination-guided FiLM.** For $\ell \geq 1$ in the decoder $G$, the synaptic current is a convolution on spikes modulated by per-channel FiLM parameters:

$$I^{(\ell)}[t] = \alpha^{(\ell)} \odot \big(W^{(\ell)} * S^{(\ell-1)}[t]\big) + \beta^{(\ell)}, \tag{9}$$

where $*$ is a standard convolution and $\alpha^{(\ell)}, \beta^{(\ell)} \in \mathbb{R}^{C_\ell}$ are time-shared (constant over $t$) and broadcast spatially. They are predicted once from temporally aggregated illumination features (from (3)):

$$\overline{F}_{\mathrm{lu}} = \frac{1}{T} \sum_{t=1}^{T} F_{\mathrm{lu}}[t], \quad \hat{f} = \mathrm{GAP}(\overline{F}_{\mathrm{lu}}), \quad [\tilde{\alpha}^{(\ell)} \| \tilde{\beta}^{(\ell)}] = W_g^{(\ell)} \hat{f} + b_g^{(\ell)}, \tag{10}$$

$$\alpha^{(\ell)} = \alpha_{\min} + (\alpha_{\max} - \alpha_{\min})\, \sigma(\tilde{\alpha}^{(\ell)}), \qquad \beta^{(\ell)} = s \tanh(\tilde{\beta}^{(\ell)}), \tag{11}$$

with $(\alpha_{\min}, \alpha_{\max}) = (0.5, 2.0)$, $s = 0.05$, and $\sigma(\cdot)$ the logistic sigmoid. When $\overline{F}_{\mathrm{lu}}$ is lower-resolution, we bilinearly upsample inside the FiLM head; BatchNorm (if used) runs in `eval` mode with EMA statistics to remain stable under sparse spikes.

**Surrogate gradients and readout.** We train end-to-end with a simple piecewise-linear surrogate derivative around the firing threshold. The network uses a single-step readout identical in training and inference:

$$Y = \mathrm{clip}\big(I_{\mathrm{lu}} + \mathrm{Head}(H^{(\mathrm{out})}[T]), 0, 1\big), \tag{12}$$

where Head is a $1 \times 1$ convolution to 3 channels; gradients through clip use a straight-through estimator on $[0, 1]$.

---

[2] A slow, causal threshold adaptation was explored for extremely dark scenes but is disabled by default to keep latency minimal.

**Objective.** The loss combines an $L_1$ image term, a firing-rate regularizer, and a temporal TV on the pre-readout state to suppress flicker without over-smoothing spatial details:

$$\mathcal{L} = \|Y - J\|_1 + \lambda_{\text{spk}} \cdot \text{AvgRate}(S) + \lambda_{\text{tv}} \cdot \text{TV}_t\big(H^{(\text{out})}[1{:}T]\big), \tag{13}$$

with $(\lambda_{\text{spk}}, \lambda_{\text{tv}}) = (10^{-4}, 5{\times}10^{-4})$.

### 3.3 Illumination-Guided Spiking Transformer

We build on the U-Net-style backbone summarized in Sec. 3.1 and Fig. 1. In this subsection we focus on how illumination guidance is injected at the decoder scales: the scale-aligned illumination features $F_{\text{lu}}$ condition both the attention (via SIGA) and FiLM modulation.

At each decoder scale, we deploy a SIGA module as the core long-range dependency unit. SIGA operates on spike trains with multi-head processing: query–key spike coincidences open hard binary gates that route value spikes, while per-head FiLM parameters—predicted from scale-aligned illumination features—modulate the $Q/K/V$ projections and gating thresholds and are shared over time. All attention computations are realized by LIF neurons unrolled across $T$ steps; BatchNorm layers (when present) run in evaluation mode with EMA statistics to maintain stable activations under spike sparsity. The feed-forward branch is a two-layer spiking MLP with LIF activations, forming a standard attention+FFN block in spiking form and serving as the decoder's illumination-aware context aggregator.

**SIGA mechanism:** Each SIGA takes spike-encoded query, key, and value streams $(Q[t], K[t], V[t])$ and a scale-aligned illumination feature $F_{\text{lu}}$. Multi-head processing is used: channels are split into $H$ heads, computed in parallel, and concatenated. For each head, $F_{\text{lu}}$ drives FiLM-style parameters that modulate the $Q/K/V$ projections and set illumination-aware gating thresholds; these parameters are predicted once and shared across time steps. Spike coincidences between $Q$ and $K$ open binary gates that route $V$ spikes, yielding an event-driven hard-attention pattern implemented by LIF neurons unrolled over $T$ steps.

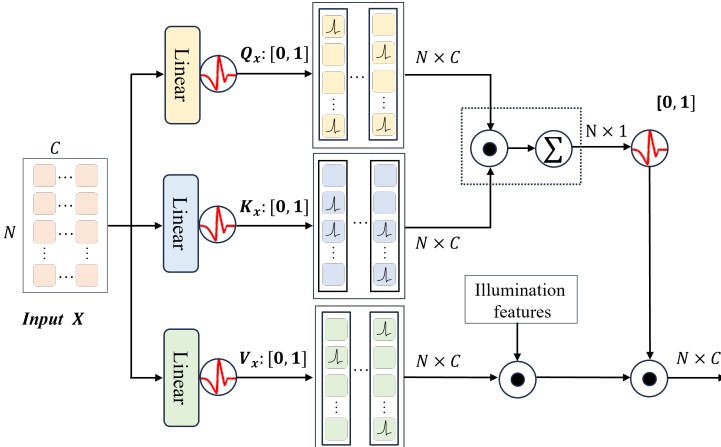

Figure 3: Schematic of the SIGA module: illustrating spike-based $Q, K, V$ interactions, modulated by the illumination features (from $\bar{L}/F_{\text{lu}}$), to generate a binary attention mask for value routing.

To compute attention in SIGA, we forego explicit floating-point matrix multiplication and softmax. Instead, spike-driven interactions implement hard gating: when a query neuron at position $i$ fires at time $t$, synapses to candidate keys are activated; if a key neuron at position $j$ simultaneously fires, the coincidence $Q_i[t]{=}1 \wedge K_j[t]{=}1$ opens an instantaneous hard gate $A_{ij}[t]$ whose threshold is modulated by illumination features. Routing uses the current $A_{ij}[t]$ (preserving causality), while optional co-firing accumulators over $t{=}1\ldots T$ integrate statistics. Owing to spike sparsity (and optionally a local neighborhood), each query selects a small subset of keys, enforcing efficient hard attention without forming dense $QK^\top$; complexity scales with spike rate and neighborhood size rather than quadratic token pairs.

Given the gates at time $t$, value spikes are routed accordingly: a value spike $V_j[t]$ contributes to query position $i$ iff $A_{ij}[t]$ is open. Implementationally, each head employs LIF neurons receiving value

| Input | UR_Retinex | EnlightenGAN | Diff-Retinex | RetinexFormer | Ours | Target |
|-------|-----------|--------------|--------------|---------------|------|--------|

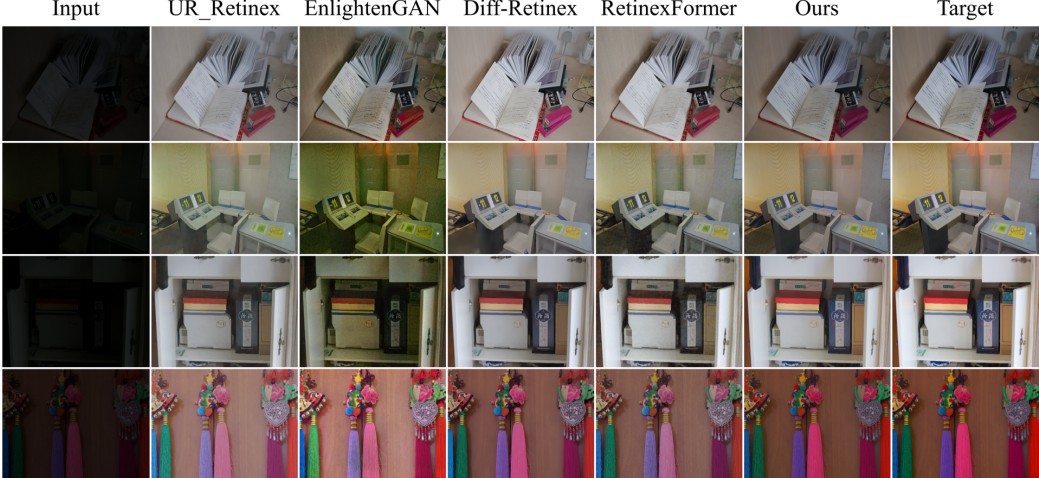

Figure 4: Low-light image enhancement results on the LOL-v1 dataset. Compared methods: URetinex-Net, EnlightenGAN, Diff-Retinex, RetinexFormer and Spike-RetinexFormer(Ours)

inputs masked by $A_{ij}[t]$; absent a gate, $V_j[t]$ has no effect on $i$. The SIGA head output at position $i$ is a spike train emitted by a LIF unit that fires when its integrated (binary-weighted) input crosses threshold; illumination features can shift this threshold via FiLM-style bias.

After the attention stage, head outputs are concatenated and passed through a spiking feed-forward module (two layers of linear spikes with an intermediate LIF activation), analogous to a Transformer FFN. Residual connections wrap the SIGA+FFN block with proper temporal alignment—residuals are added in the membrane-potential domain at each time step—so that the training–inference readout remains identical to the rest of the network. Multiple SIGA blocks are placed across decoder stages; at each scale, illumination-guided spiking attention supplies long-range, lighting-aware context while preserving event-driven efficiency.

# 4 Experiments

## 4.1 Experimental Setup

We evaluate Spike-RetinexFormer on a comprehensive suite of standard benchmarks for LLIE, largely following the protocol of [43]. These include: LOL-v1 [6], LOL-v2 (real and synthetic) [44], SID [45], SMID [46], SDSD (indoor and outdoor) [47], MIT-Adobe FiveK [48], and LIME[49] which commonly used for qualitative evaluation. For datasets with paired ground truth, we train a separate model per dataset using the official or widely adopted splits; for RAW datasets, we convert to sRGB via the standard ISP pipeline before computing losses and metrics to ensure comparability.

All experiments share the same backbone and training hyperparameters as in Sec. 3. Unless otherwise stated, we use the AdamW optimizer (base learning rate $2 \times 10^{-4}$) with cosine annealing (optional warm-up), unrolling the spiking dynamics for $T$ time steps and applying global-norm clipping at 1.0. Each model is trained until the validation performance saturates. We report the Peak Signal-to-Noise Ratio (PSNR) and Structural Similarity (SSIM) [50] on sRGB results for quantitative comparison. For model complexity, we provide parameter counts (Millions) and FLOPs at $256 \times 256$ input resolution.

## 4.2 Comparison with ANN models

We compare Spike-RetinexFormer with recent ANN methods, including Retinex-based CNNs and Transformer-based models: RetinexNet [6], KinD [51], DeepUPE [52], RUAS [53], MIRNet [54], Uformer [55], Restormer [26], SNR-Net [56], and RetinexFormer [43]. Quantitative results on benchmarks with ground truth are summarized in Tab. 1 (LOL) and Tab. 2 (SID, SMID, SDSD). Overall, Spike-RetinexFormer delivers competitive performance across most datasets and often matches strong ANN baselines. Notably, on the widely used LOL-v1 dataset, our method attains

Table 1: Quantitative comparison on the datasets of LOL-v1, LOL-v2 (real and synthetic).

| Method | FLOPs (G) | Params (M) | LOL-v1 PSNR/SSIM | LOL-v2-R PSNR/SSIM | LOL-v2-S PSNR/SSIM |
|---|---|---|---|---|---|
| RetinexNet [6] | 587.5 | 0.84 | 16.77/0.560 | 15.47/0.567 | 17.13/0.798 |
| DeepUPE [52] | 21.1 | 1.02 | 14.38/0.446 | 13.27/0.452 | 15.08/0.623 |
| KinD [51] | 35.0 | 8.02 | 20.86/0.790 | 14.74/0.641 | 13.29/0.578 |
| RUAS [53] | 0.8 | 0.003 | 18.23/0.720 | 18.37/0.723 | 16.55/0.652 |
| MIRNet [54] | 785.0 | 31.60 | 24.14/0.830 | 20.02/0.820 | 21.94/0.876 |
| Restormer [26] | 144.3 | 26.10 | 22.43/0.823 | 19.94/0.827 | 21.41/0.830 |
| SNR-Net [56] | 26.3 | 4.01 | 24.61/0.842 | 21.48/0.849 | 24.14/0.928 |
| RetinexFormer [43] | 15.6 | 1.61 | 25.16/0.845 | 22.80/0.840 | 25.67/0.930 |
| Ours | 16.2 | 1.50 | 25.50/0.842 | 23.38/0.848 | 26.47/0.938 |

Table 2: Quantitative comparison on SID, SMID, and SDSD (indoor/outdoor) datasets. Our method achieves top performance across these diverse benchmarks.

| Method | SID PSNR/SSIM | SMID PSNR/SSIM | SDSD-in PSNR/SSIM | SDSD-out PSNR/SSIM | FLOPs (G) | Params (M) |
|---|---|---|---|---|---|---|
| KinD [51] | 18.02/0.583 | 22.18/0.634 | 21.95/0.672 | 21.97/0.654 | 35.0 | 8.02 |
| MIRNet [54] | 20.84/0.605 | 25.66/0.762 | 24.38/0.864 | 27.13/0.837 | 785.0 | 31.60 |
| Uformer [55] | 18.54/0.577 | 27.20/0.792 | 23.17/0.859 | 23.85/0.748 | 12.0 | 5.29 |
| Restormer [26] | 22.27/0.649 | 26.97/0.758 | 25.67/0.827 | 24.79/0.802 | 144.3 | 26.10 |
| SNR-Net [56] | 22.87/0.625 | 28.49/0.805 | 29.44/0.894 | 28.66/0.866 | 26.3 | 4.01 |
| RetinexFormer [43] | 24.44/0.680 | 29.15/0.815 | 29.77/0.896 | 29.84/0.877 | 15.6 | 1.61 |
| **Ours** | **24.68/0.681** | **29.43/0.820** | **30.45/0.903** | **30.37/0.885** | **16.2** | **1.50** |

25.5 dB PSNR and 0.842 SSIM, which is on par with RetinexFormer (25.2 dB, 0.845) and clearly ahead of earlier Retinex-based CNNs such as RetinexNet (16.8 dB) and KinD (20.8 dB). On the more challenging LOL-v2 (synthetic), Spike-RetinexFormer reaches 26.5 dB PSNR, representing a 0.8 dB gain over RetinexFormer with comparable SSIM; trends on SID/SMID/SDSD are similar. We attribute this competitiveness to the synergy between Retinex decomposition and spiking neural dynamics: illumination-guided spiking attention helps balance contrast, while iterative spike integration aids noise suppression under extreme low light. In terms of perceptual quality, our results typically exhibit fewer color shifts and artifacts; as shown in Fig. 4, our method yields natural, well-exposed images with preserved details, whereas some baselines may over-smooth or leave residual noise.

As seen in Tab. 1 and 2, our spiking approach shows competitive or improved performance across the LOL variants and the extremely dark, noisy datasets (SID, SMID, SDSD). On average, Spike-RetinexFormer improves PSNR by $\sim 0.5$ dB over strong prior methods, with gains up to $\sim 0.8$ dB depending on the benchmark. In particular, for the dark indoor scenes of SDSD, our method recovers additional details, reaching 30.45 dB. Meanwhile, the model is compact—1.5M parameters and 16.2 G FLOPs—noticeably lower than heavy Transformers such as Restormer (26M, 144G). We attribute this efficiency to the one-stage design and the sparsity of spiking computation (neurons do not fire at every time step), making the approach practical for deployment. On neuromorphic hardware, energy consumption is expected to decrease because operations are triggered by spikes rather than dense activations. For example, if a 32-bit MAC costs an order of magnitude more energy than an accumulate (AC) [19], and only $\sim 18\%$ of neurons fire per time step in our network (Tab. 5), a back-of-the-envelope estimate suggests on-the-order-of $5\times$–$10\times$ energy reduction versus comparable ANN models under these assumptions. While these savings are theoretical (we have not measured power on a neuromorphic chip), they highlight the potential advantage of event-driven processing for power efficiency.

### 4.3 Ablation Studies

We systematically ablate the one-stage Retinex formulation (ORF), the proposed SIGA, and the influence of the time step $T$. All ablations are conducted on LOL-v1 using a single RTX 3090.

Table 3: Backbone ablations on LOL-v1 (full-resolution).

| Variant | PSNR | SSIM | Params (M) | FLOPs (G) |
|---|---|---|---|---|
| Baseline (Spike-U-Net) | 23.11 | 0.789 | 1.20 | 9.5 |
| + ORF | 24.56 | 0.822 | 1.45 | 11.8 |
| + ORF + W-MSA | 25.07 | 0.835 | 1.62 | 13.2 |
| + ORF + G-MSA | 25.29 | 0.837 | 1.75 | 16.8 |
| **+ ORF + SIGA (ours)** | **25.50** | **0.842** | 1.68 | 14.0 |

Table 4: Ablation on neuron type and architecture (LOL-v1, $T=8$).

| Activation | Architecture | PSNR | SSIM | Energy (mJ) |
|---|---|---|---|---|
| ReLU | RetinexFormer | 25.16 | 0.845 | 71.6 |
| LIF | RetinexFormer | 20.31 | 0.714 | 7.5 |
| LIF | Spike-RetinexFormer | 25.50 | 0.842 | 16.7 |
| IF | Spike-RetinexFormer | 25.31 | 0.831 | 18.9 |
| PLIF [57] | Spike-RetinexFormer | 25.49 | 0.833 | 15.8 |
| CLIF [58] | Spike-RetinexFormer | 25.61 | 0.848 | 16.3 |

Tab. 3 compares five progressively enhanced variants. **Baseline** is a pure spiking U-Net (no illumination branch, no SIGA). Introducing the illumination estimator and light-up operation (+ ORF) yields a +1.45 dB PSNR gain (23.11→24.56), indicating that explicit exposure modeling is critical in the spiking regime. Augmenting the ORF backbone with either a local-window MSA (+ W-MSA) or a global MSA (+ G-MSA) brings additional improvements. Our SIGA attains the highest fidelity, outperforming W-MSA and G-MSA by +0.43 and +0.21 dB. Qualitatively, the advantage is most evident in extreme shadows and mixed-lighting regions, where illumination-aware gating improves detail recovery and noise suppression.

We ablate neuron types on LOL-v1 ($T=8$) with: (i) ANN RetinexFormer; (ii) RetinexFormer with LIF (ReLU→LIF); (iii) Spike-RetinexFormer (LIF+ORF+SIGA); and (iv) the same spiking backbone with IF/PLIF/CLIF (Tab. 4). Swapping ReLU→LIF in the ANN cuts compute energy to 7.5 mJ (10.5% of baseline) but reduces fidelity (PSNR −4.85 dB; SSIM 0.845 → 0.714). Adding ORF+SIGA in the full spiking model restores accuracy while keeping energy low: PSNR 25.50 dB (+0.34 dB vs ANN), SSIM ≈ ANN, and 16.7 mJ (down from 71.6 mJ, −77%). Fixing the architecture, neuron choice yields small but consistent gaps: CLIF is best (25.61/0.848) at energy close to LIF; IF/PLIF are slightly lower, 15.8–18.9 mJ.

Tab. 5 reports performance and efficiency under varying time-step configurations. Using fewer steps ($T=4$) provides limited temporal evidence, leading to a modest PSNR drop and a higher average spike firing rate (SFR). Increasing to $T=8$ achieves a favorable accuracy–latency balance; pushing to $T=12$ offers only

Table 5: Impact of time steps $T$. Avg. SFR is the percentage of active neurons per step; latency is wall-time on RTX 3090 (ms); $\Delta$Energy is normalized to $T=8$.

| $T$ | PSNR | SSIM | Avg. SFR | Latency | $\Delta$Energy |
|---|---|---|---|---|---|
| 4 | 25.21 | 0.836 | 0.238 | 16.5 | 0.55× |
| 8 | 25.50 | 0.842 | 0.187 | 31.0 | 1.00× |
| 12 | 25.59 | 0.841 | 0.185 | 46.2 | 1.32× |

marginal PSNR gains while increasing wall-time nearly linearly. We therefore adopt $T=8$ as the default trade-off.

# 5   Limitations and Future Work

**Hardware-validated efficiency and scalability.** Although SIGA avoids dense softmax attention, computing spike co-firing statistics can still scale as $O(N^2)$ in the number of spatial tokens $N$ when firing rates increase or sparsity collapses in high-illumination regions. To address this, we will design an event-driven sparse attention kernel that constructs co-firing pairs from time-bucketed inverted indices, enumerating only observed spike events and thereby avoiding any dense map materialization. In addition, we will investigate top-$k$ gating, block/tile-level sparsity, and kernel-level fusion of FiLM-style illumination modulation to reduce compute, memory traffic, and latency, and will release kernels and power traces to facilitate reproducibility.

**Robustness and cross-domain generalization.** We will pursue RAW-aware training via a differentiable imaging pipeline and camera-conditional normalization to account for sensor variability; cross-dataset adaptation with self-supervised consistency constraints across RAW–sRGB pairs; and uncertainty-aware exposure control that disentangles aleatoric and epistemic components to mitigate over- and under-enhancement. We also plan to extend the framework to video using temporal spike-consistency losses and frame-adaptive time steps, and to broaden evaluation with no-reference and perceptual metrics (NIQE, BRISQUE, LPIPS) alongside stress tests for flicker, haloing, and color fidelity to more rigorously assess deployment robustness.

## 6 Conclusion

We introduced Spike-RetinexFormer, a low-light image enhancer that unifies spiking neural networks with a Retinex-inspired Transformer architecture. By guiding spiking self-attention with an estimated illumination map, our approach achieves competitive enhancement quality on challenging dark images while demonstrating promising energy efficiency and compact memory use via event-driven, sparse computation. By integrating the computational efficiency of SNNs with the representational strengths of Transformers for low-light enhancement, Spike-RetinexFormer offers a practical blueprint for high-performance, energy-frugal vision systems on power-constrained platforms and points toward hardware-validated evaluation and video extensions.

## 7 Acknowledgement

This work was partly supported by Ningbo Youth Science and Technology Innovation Leading Talent Project (2024QL044) and Ningbo Key R&D Program (2025Z047).

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
