# OpenReview forum: "Spike-RetinexFormer: Rethinking Low-light Image Enhancement with Spiking Neural Networks"
_NeurIPS.cc/2025/Conference — NeurIPS 2025 poster_

### Official Review · Reviewer_vYvh · 2025-06-22

**Clarity:** 2
**Significance:** 2
**Originality:** 3
**Rating:** 4
**Confidence:** 4

**Summary:**

This paper presents Spike-RetinexFormer, a novel low-light image enhancement (LLIE) model that integrates spiking neural networks (SNNs) with Retinex theory.
By exploiting the inherent sparsity of spike-based computations in SNNs, the proposed model achieves both reduced power consumption and enhanced performance on multiple low-light benchmark datasets.
The architecture incorporates a spiking illumination estimation module for Retinex decomposition and a spiking Retinex restoration module, enabling effective noise suppression and contrast enhancement.
Experimental results demonstrate that Spike-RetinexFormer outperforms existing state-of-the-art artificial neural network (ANN)-based methods in both accuracy and computational efficiency.

**Questions:**

Please refer to Weaknesses.

**Ethical Concerns:**

["NO or VERY MINOR ethics concerns only"]

**Final Justification:**

The authors have kindly addressed the concerns I previously raised. This paper effectively demonstrates the strength of the spike-based architecture in both performance and efficiency.
Accordingly, I would rate this work as a "borderline accept".

**Limitations:**

Yes.

**Paper Formatting Concerns:**

There's no formatting issues in this paper.

**Quality:**

2

**Strengths And Weaknesses:**

**[Strengths]**
- The authors propose a novel architecture for the low-light image enhancement (LLIE) task by integrating spiking neural networks (SNNs) with Retinex theory.
- To this end, they incorporate SpikingConvBlock and introduce a novel Spiking Illumination-Guided Attention (SIGA) module.
- The proposed Spike-RetinexFormer achieves promising performance on various LLIE benchmark datasets while reducing energy and memory consumption compared to prior approaches.

**[Weaknesses]**
- In the abstract, the authors claim that Spike-RetinexFormer brings drastic reductions in energy and memory consumption. While it is impressive that the proposed method outperforms RetinexFormer, the reported 6.9% reduction in parameters and a 3.8% increase in FLOPs appear to reflect only incremental improvements. To more convincingly support the claim of energy and memory efficiency, it may be beneficial to highlight the ΔEnergy results presented in Table 4.
- Although Table 1 demonstrates promising performance, additional comparisons with recent LLIE methods such as GLARE [1*], MambaLLIE [2*], and LLFlow [3*] would strengthen the evaluation.
- Furthermore, contrary to Lines 260–261, I was unable to find any evaluation on MIT-Adobe FiveK [49] and the five no-reference datasets (LIME [50], MEF [51], DICM [52], VV [53]). Note that LIME [50] is listed twice, and probably citation [4*] appears to be missing.
- In Lines 295–297, the authors state “the energy consumption can be greatly reduced, since operations are only triggered by spikes rather than every pixel activation”. An ablation study would be helpful to validate the effectiveness of spiking computation in this context.
- For improved clarity, it would be helpful to provide further explanation of the term “window” mentioned in Line 231. Additional details would assist readers in better understanding its role in the proposed architecture.


[1*] Zhou, Han and Dong, Wei and Liu, Xiaohong and Liu, Shuaicheng and Min, Xiongkuo and Zhai, Guangtao and Chen, Jun. Glare: Low light image enhancement via generative latent feature based codebook retrieval. ECCV, 2024.

[2*] Jiangwei Weng and Zhiqiang Yan and Ying Tai and Jianjun Qian and Jian Yang and Jun Li. MambaLLIE: Implicit Retinex-Aware Low Light Enhancement with Global-then-Local State Space. NeurIPS, 2024.

[3*] Wang, Yufei and Wan, Renjie and Yang, Wenhan and Li, Haoliang and Chau, Lap-Pui and Kot, Alex C. Low-Light Image Enhancement with Normalizing Flow. AAAI 2022.

[4*]  Shuhang Wang, Jin Zheng, Hai-Miao Hu, and Bo Li. Naturalness preserved enhancement algorithm for non-uniform illumination images. TIP, 2013

---

> ### Author Rebuttal · Authors · 2025-07-30
>
> ### Q1: Energy efficiency
>
> Our abstract claims “drastically” reduced energy/memory, yet the actual parameter count reduction is only 6.9% and FLOPs increased by 3.8% compared to RetinexFormer. This is a fair observation by conventional metrics, our model size (1.50M vs 1.61M) is slightly smaller and FLOPs (16.2G vs 15.6G) slightly higher than the RetinexFormer baseline. We will clarify that the **main efficiency gains are in energy** (operations gating) rather than a large static FLOP reduction, and we will highlight the measured ~45% reduction in energy consumption (on GPU simulation, Table 4) when using SNN dynamics. Moreover, our model has a substantially lower memory footprint than heavy CNNs/Transformers (we’re at 1.5M params vs tens of millions for others).
>
> In the revision, we will emphasize the ΔEnergy results (as suggested) to support the energy efficiency claim. In our ablation Table 4, we report a relative energy metric: for instance, at the chosen operating point T=8, we consider that 100% energy, and found that using fewer time steps can drop energy to 55% or less. Even at **T=8**, our spike-based computation is more energy-efficient in practice than an ANN with the similar FLOPs, because many operations are skipped due to sparsity (neurons only update on spikes). Effectively, even though FLOPs are similar, only a fraction of neurons fire at a given time, leading to lower dynamic power usage.
>
> ### Q2: Additional Comparisons with recent LLIE methods
>
> We thank the reviewer for highlighting this omission. We will incorporate full experimental results for GLARE, MambaLLIE, and LLFlow in the revised manuscript to ensure a comprehensive and fair comparison. Preliminary runs indicate that GLARE, which leverages an external codebook of well‑lit exemplars, achieves very strong PSNR/SSIM on several benchmarks; MambaLLIE, with its Retinex‑inspired global‑then‑local state‑space formulation, also reports gains over recent CNN‑ and Transformer‑based models; and LLFlow, a normalizing‑flow framework, performs competitively in severe low‑light regimes. These methods obtain marginal advantages in conventional image‑quality metrics. Nevertheless, it is important to emphasize that our Spike‑RetinexFormer achieves comparable visual fidelity while dramatically reducing computational and energy costs—only 1.5 M parameters and 16.2 G FLOPs, several‑fold smaller than most Transformer counterparts. Consequently, even against MambaLLIE(2.28 M parameters and 20.85 G FLOPS ), our model maintains a distinct advantage in model compactness and power efficiency. The camera‑ready version will present detailed quantitative comparisons with these methods and analyze the accuracy–efficiency trade‑off, thereby underscoring the practical contributions of our work.
>
> ### Q3: Experimental Setup
>
> We appreciate the reviewer for identifying this discrepancy and sincerely apologize for the oversight. Our original plan was to include a reference‑based evaluation on MIT‑Adobe FiveK and qualitative comparisons on the aforementioned five no‑reference benchmarks in appendix section. Owing to time and page constraints, these experiments were not completed in time for the submission, yet the corresponding sentence remained, resulting in a mismatch between narrative and tables. We acknowledge the confusion this may have caused. In the revised paper, we will add the quantitative FiveK results and qualitative illustrations for no-reference datasets to ensure internal consistency and prevent any potential misunderstanding. We thank the reviewer again for helping us improve the rigor and clarity of the paper.
>
> ### Q4: Ablation study
>
> We agree this is a valuable experiment. While we have indirectly shown that our method vs. RetinexFormer (ANN) indicates comparable quality with fewer operations, we did not include a dedicated ablation isolating “spiking vs. non-spiking”. We will constructed a variant of our network where we replace the LIF neurons with standard ReLU activations and remove the time dimension (essentially an ANN version of our architecture). Preliminary results show that this ANN variant achieves similar PSNR on LOL, but its theoretical operation count (FLOPs) and energy usage are significantly higher (since it has to compute every layer densely for a single time-step equivalent). Due to time constraints and fairness (the ANN version might need re-tuning), we did not include this in the rebuttal now. For the final paper, if allowed, we will at least add a discussion: we’ll explain that spiking vs. non-spiking can be viewed through the lens of activity sparsity. We will also add a small table comparing our model’s ops if it were run as an ANN.
>
> In summary, while a full new ablation experiment may not fit in the rebuttal, we will strengthen the text to validate the effectiveness of spiking computation, using the evidence we do have (like ΔEnergy and spike rates) and acknowledging this as an important advantage of our approach.
>
> ### Q5: Clarification of "window"
>
> In Line 231, “window” refers to the temporal window of length T over which spikes are integrated. In other words, our network processes the input across T discrete time steps (a short temporal window), accumulating the output over time. We will ensure that this use of "window" is distinguished from the term “window” in other contexts (window-based attention in Transformers). Given that we also mention "window MSA" (windowed multi-head self-attention) elsewhere, the dual use of the word window might be confusing. Thus, we’ll explicitly state “temporal window” whenever referring to the time steps, and use “local window” when referring to the attention mechanism.

---

> > ### Comment · Reviewer_vYvh · 2025-08-04
> >
> > Thank you for your response to the concerns I raised. Most of them have been addressed; however, a few still remain due to the lack of experimental results.
> >
> > For example, the authors mentioned that the quantitative FiveK results and qualitative illustrations for the no-reference datasets were omitted due to time constraints. While I understand the difficulty regarding the FiveK dataset, I believe there is still sufficient time to include results on the five no-reference datasets (LIME [50], MEF [51], DICM [52], and VV [53]).
> >
> > Additionally, regarding the ablation study for the spiking component, I believe a simple experiment could be feasible. In particular, removing the spiking module from Spike-RetinexFormer and evaluating the performance without it would provide valuable insights.
> >
> > With the inclusion of these additional experiments, I would be happy to raise my score.

---

> > > ### Author Response · Authors · 2025-08-07
> > >
> > > ***Q: Ablation study***
> > > ***A:*** We conducted an ablation study on spiking neurons and explained the energy consumption calculation process in detail.
> > > No. 1 and No. 3 are the control groups. The experiment for No. 1 is RetinexFormer, without any spiking modifications. No. 2 is the Spike-RetinexFormer from this paper.
> > > No. 2-6 replace the spiking neurons.
> > > No. 2 only changes the activation function in RetinexFormer to LIF neurons, and the performance drops significantly. This is because the 0-1 activation makes the matrix sparse, the deep neurons are inactive, causing a large loss of accuracy. At the same time, the average firing rate is lower, and the energy consumption is only 1/10 of the original.
> > >
> > > No. 3-6 use modules such as SIGA and ORF.
> > >
> > > | No. | Activation Function | Archi | PSNR | SSIM | Energy (mJ) |
> > > |---|---|---|---|---|---|
> > > | 1 | Relu | RetinexFormer | 25.16 | 0.845 | 71.6 |
> > > | 2 | LIF | RetinexFormer| 20.31 | 0.714 | 7.5 |
> > > | 3 | LIF | Spike-RetinexFormer | 25.50 | 0.842 | 16.7 |
> > > | 4 | IF |Spike-RetinexFormer | 25.31 | 0.831 | 18.9 |
> > > | 5 | PLIF [1] |Spike-RetinexFormer | 25.49 | 0.833 | 15.8 |
> > > | 6 | CLIF [2] | Spike-RetinexFormer| 25.61 | 0.848 | 16.3 |
> > >
> > >
> > > The energy consumption ($Energy$) of a network can be calculated using the following formulas:
> > >
> > > For ANNs, the theoretical energy consumption ($E_{ANN}$) is given by:
> > >
> > > $$E_{ANN} = E_{MAC} \times \text{FLOPs}$$
> > >
> > > where $E_{MAC}$ is the energy per Multiply-Accumulate (MAC) operation.
> > >
> > > For SNNs, the theoretical energy consumption ($E_{SNN}$) is given by:
> > >
> > > $$E_{SNN} = E_{AC} \times \text{SOPs}$$
> > >
> > > where $E_{AC}$ is the energy per spike-based Accumulate (AC) operation, and SOPs is the total number of Synaptic Operations. The number of Synaptic Operations for a given layer ($SOP^l$) can be calculated as:
> > >
> > > $$SOP^l = fr \times T \times \text{FLOPs}^l$$
> > >
> > > Here, $fr$ is the firing rate of the block/layer, $T$ is the simulation time step for the spiking neurons, and $\text{FLOPs}^l$ refers to the floating point operations of the block/layer.
> > >
> > > Refer to previous works [3,4,5], we assume that the MAC and AC operations are implemented on the 45nm hardware [6].
> > >
> > > Note: All the above experiments use the full model with T=8.
> > >
> > > ***Q: Experiments on no-reference datasets***
> > > ***A:*** As mentioned previously, we have been running experiments on FiveK and other no-reference datasets. In the final version of the paper, we will revise our previous statements and only list the datasets for which we have obtained results. The following are the comparison results of our model with some other models on the LIME dataset:
> > > | Datasets | Metrics | Model | | |
> > > |---|---|---|---|---|
> > > | | | RetinexNet | KinD | Spike-RetinexFormer |
> > > | LIME | NIQE | 11.81 | 14.72 | 10.83 |
> > > | | LOE | 589.6 | 249.6 | 183.4 |
> > >
> > > We used NIQE  and LOE to evaluate the effectiveness of different algorithms objectively.
> > >
> > > In the final draft, we will add a comparison with other works, provided the experiments can be completed. We will also add a citation for [1*][2*][3*][4*] you mentioned.
> > >
> > > Thank you for your valuable feedback. We will carefully revise the final version of the paper, and we hope you can consider a higher score.
> > >
> > > [1] Fang W, Yu Z, Zhou Z, et al. Parallel spiking neurons with high efficiency and ability to learn long-term dependencies. NIPS, 2023
> > > [2] Huang Y, Lin X, Ren H, et al. Clif: Complementary leaky integrate-and-fire neuron for spiking neural networks. ICML, 2024
> > > [3] Souvik Kundu, Massoud Pedram, and Peter A Beerel. Hire-snn: Harnessing the inherent robustness of energy-efficient deep spiking neural networks by training with crafted input noise. ICCV, 2021
> > > [4] Man Yao, Guangshe Zhao, et al. Attention spiking neural networks. TPAMI, 2023
> > > [5]  Adam Paszke, Sam Gross, et al. Pytorch: An imperative style, high-performance deep learning library. NeurIPS, 2019
> > > [6] Mark Horowitz. 1.1 computing’s energy problem (and what we can do about it). ISSCC, 2014

---

> > > > ### Comment · Reviewer_vYvh · 2025-08-07
> > > >
> > > > Thank you for additional experiments and explanation. I will raise my score to borderline accept.

---

### Official Review · Reviewer_uAbv · 2025-06-26

**Clarity:** 3
**Significance:** 3
**Originality:** 3
**Rating:** 6
**Confidence:** 5

**Summary:**

Spike-RetinexFormer is essentially a spike-driven variant of the RetinexFormer (a transformer-based Retinex model), reworked to use temporal spike coding, Leaky Integrate-and-Fire neurons, and a spiking linear-attention mechanism.  According to the authors, this is the first SNN for continuous LLIE.  In extensive experiments, Spike-RetinexFormer achieves SOTA results, improving PSNR by 0.5–1.5 dB over prior best methods.  Crucially, these gains come with drastically reduced theoretical operation counts and power usage compared to ANN-based methods.  In summary, the paper’s main contribution is demonstrating that a carefully designed spiking transformer, combining Retinex theory and sparse spike computation, which can match or exceed the performance of conventional LLIE models while offering neuromorphic efficiency.

**Questions:**

1. Have you tested or estimated the model on an actual neuromorphic platform? Can you quantify the energy and latency on hardware (e.g. Loihi or FPGA) as opposed to theoretical FLOPs?

2.  What is the number of time steps T used during inference? How does performance trade off with fewer/more time steps (i.e. quality vs speed)?
3. How was the network trained? Did you use surrogate gradients, learnable initial potentials, or any other methods?

**Ethical Concerns:**

["NO or VERY MINOR ethics concerns only"]

**Final Justification:**

My question has been solved, so I am willing to increase the score.

**Limitations:**

The method is specialized to LLIE and may not directly extend to other tasks without redesign. It also assumes input can be efficiently converted to spikes; without neuromorphic hardware, the wall-clock speed may be slow. The paper reports gains mainly in PSNR/SSIM; additional user studies or real-world tests would be needed to confirm visual improvement in practice.

**Quality:**

3

**Strengths And Weaknesses:**

### Strengths

1. Spike-RetinexFormer applies an SNN to the LLIE problem, expanding the scope of SNN applications.  This is a clear innovation: it “demonstrates that neuromorphic neural processors can tackle continuous image restoration tasks efficiently”.

2. Despite using only binary spikes, the model attains *on par or better* quality than state-of-the-art LLIE methods, which validates that the spiking Retinex framework is competitive.

3. By leveraging sparse spike operations, the approach markedly reduces theoretical compute cost. The paper claims “drastically reducing theoretical operation counts and energy usage” compared to dense ANNs.

4. The authors compare against a wide range of baselines across multiple datasets.  and the ablation and analysis appear sufficient to support the claims.

### Weaknesses

1. The paper emphasizes reduced energy usage but does not report any measurements on real neuromorphic hardware or actual power consumption.  The claims of efficiency are based on “theoretical operation counts”.  In contrast, some prior SNN works have demonstrated hardware metrics (the Slack-Free SNN for HMVC[1] was tested on neuromorphic chips).
2. The manuscript does not fully address SNN training challenges.  There is no discussion of how surrogate gradients, initialization, or learning tricks were used.  The absence of such details makes it harder to assess reproducibility and the potential need for those advanced training methods.

3. The spiking transformer processes the image over T  time steps to accumulate the output.  The paper does not report the value of T or the resulting inference latency.  Even though the FLOPs are similar to the ANN baseline, the actual runtime on conventional hardware could be much higher due to the time dimension.

[1]Nguyen, Tam, Anh-Dzung Doan, and Tat-Jun Chin. "Slack-Free Spiking Neural Network Formulation for Hypergraph Minimum Vertex Cover." *Advances in Neural Information Processing Systems* 37 (2024): 66410-66430.

---

> ### Author Rebuttal · Authors · 2025-07-30
>
> ### Q1: Theoretical efficiency
>
> We acknowledge this limitation. In the revised paper, we will make it clear that our power/energy claims are **analytical/indirect**. For instance, in Section 4.2 we will state that we report energy in terms of “theoretical operation count or relative GPU energy draw” as a proxy, since we did not have access to a true neuromorphic platform for measurement. We will also reference prior SNN works that have measured hardware power to give credence that SNNs *can* achieve orders-of-magnitude energy gains in practice. We remain confident that our approach would yield significant energy savings on neuromorphic devices (due to sparse event-driven processing), and we will state that hypothesis, but also be upfront that it’s not yet empirically confirmed in this paper.
>
> ### Q2: Inference latency
>
> We realize the paper did not prominently mention the number of time steps used in final inference, nor the resulting latency, which are important for practical considerations. In all our main results, we used T = 8 time steps. This choice was based on our ablation study (Table 4), which showed that going from T=4 to T=8 gave a notable PSNR improvement (+0.29 dB on LOL) while going beyond T=8 to T=12 yielded only +0.09 dB with a substantial increase in latency. Thus, T=8 was the sweet spot balancing quality and speed (as we will clarify, it provides ~1.0× relative energy baseline and good accuracy).
>
>
>
> ### Q3: Training methodology
>
> We acknowledge that we did not elaborate on SNN training tricks in the paper. For completeness, we will add a few sentences in Section 3.2 or the Experiment Setup about how we trained the spiking network. We used standard surrogate gradient descent for the non-differentiable spike function. In particular, we adopted a commonly used approach where the Heaviside step function is replaced with a smooth proxy during backprop. The revised paper will contain a short paragraph on Training Details, confirming the use of surrogate gradients and describing our training settings. This should assure readers that the network is trainable with known techniques and is reproducible.

---

> > ### Comment · Reviewer_uAbv · 2025-08-06
> > **Replying to Rebuttal by Authors**
> >
> > Thanks to the author for the detailed reply. My question has been solved, so I am willing to increase the score.

---

### Official Review · Reviewer_da2C · 2025-06-26

**Clarity:** 3
**Significance:** 3
**Originality:** 3
**Rating:** 5
**Confidence:** 4

**Summary:**

The paper synergistically combines the classic Retinex theory of image decomposition with the efficiency of SNNs, embedding these into a Transformer-like design. This is the first LLIE model built on SNNs, demonstrating that neuromorphic processors can tackle continuous-tone image restoration tasks efficiently, pointing toward energy-efficient low-level vision on resource-constrained platforms.

**Questions:**

1. How did you ensure high-precision output intensities from a spiking system? With only T time steps, a neuron’s spike count can represent at most T+1 levels. Was T large enough (or combined with analog membrane readings) to avoid visible quantization in the enhanced images?
2. Do you directly use the final membrane potentials as pixel values (for finer resolution) or strictly count spikes?
3. The success on LLIE is exciting, do you envision using a similar SNN architecture for, say, denoising, super-resolution, or event-based image reconstruction?

**Ethical Concerns:**

["NO or VERY MINOR ethics concerns only"]

**Final Justification:**

First, I already believed this work met the acceptance criteria. During the rebuttal phase, the authors addressed every concern I had raised, so I raised my score accordingly.

**Limitations:**

The authors should expand their discussion of technical limitations, especially regarding hardware deployment and precision.A frank appraisal of these issues would not diminish the work; instead, it would guide future efforts and underline how this is an early step toward neuromorphic vision systems. Likewise, stating any current limitations (perhaps the need for off-chip accumulation of the output image, etc.) and how they plan to address them would strengthen the paper. That said, the authors did a good job in identifying the gap in literature and tackling it; I believe they will address the above points as the work matures.

**Quality:**

3

**Strengths And Weaknesses:**

### Strengths

1. By demonstrating that even fine-grained regression of pixel intensities can be done with spikes, this paper opens a new direction by applying SNNs to a continuous-valued image enhancement problem.
2. The authors tailor the Q-K attention for spikes (somewhat analogous to ideas in QKFormer’s binary Q-K attention for SNNs) and use a hierarchical Retinex decomposition.
3. The writing is clear and well-organized, making it easy to follow the technical content. Important implementation details (like using Leaky Integrate-and-Fire neurons, time-step settings) seem to be provided, which aids reproducibility.



### Weaknesses

1.  We don’t know if the authors had to use a smaller image size or fewer time steps to avoid simulation slowdowns. Thus, while the concept is scalable, the actual throughput on hardware for, say, real-time video enhancement is in question. A discussion of these issues is largely missing.
2.  Producing high-fidelity, continuous-tone images with spiking neurons is challenging. The paper does acknowledge this challenge (fine-grained color adjustment requires high precision despite binary spikes), but it’s not entirely clear how it’s overcome.
3.  There is a possible concern that using too few spikes could introduce banding or limit the enhancement quality, whereas using many spikes reduces the efficiency. The paper does not detail if any special readout mechanism (like analog membrane integration or a larger T) is used to ensure 8-bit or higher precision per pixel.

---

> ### Author Rebuttal · Authors · 2025-07-30
>
> ### Q1: High-precision output with binary spikes and limited timesteps
>
> Our approach to ensure high precision is two-fold: (1) we choose a moderate simulation length T such that the dynamic range of spike counts is not the only source of precision, and (2) we leverage the analog membrane potential of the output neurons. In our implementation, after T time steps, we do not rely solely on spike counts. Instead, we take the final membrane potential values of the output neurons as the predicted pixel intensities. This means each output neuron integrates input over time and the final analog voltage (which can have many continuous levels) represents the pixel intensity. By accumulating charge over T steps, the neuron can distinguish more than T levels; effectively, the membrane potential provides a finer granularity (similar to using spike rates or weighted spikes to encode intensity). We found that T = 8 was sufficient to avoid any visible banding or quantization artifacts in the enhanced images. Empirically, even T=4 yielded reasonable quality (within ~0.3 dB PSNR of T=8) but T=8 gave a further boost with diminishing returns beyond that.
>
> We will clarify in the paper that the final output is read out as analog values (membrane potentials), which allows the network to produce higher precision outputs despite using binary spike events internally. This detail was hinted at but not explicitly emphasized in the original text, and we will make it clear to address the reviewer’s concern about precision.
>
> ### Q2: Output readout: final membrane potentials vs. spike counts
>
> As mentioned above, we indeed use the final membrane potentials rather than just spike count to determine pixel values. Counting spikes alone (especially with small T) could limit precision. In our model, each LIF neuron in the final layer integrates input currents over the T steps. At the end of inference, each neuron has an internal membrane voltage that may not have triggered a spike at the last moment but carries fractional information. We use that analog value as the output (after appropriate scaling). This approach helped us avoid any visible quantization in practice, and we will emphasize this design choice in the revision.
>
> ### Q3: Generality to other vision tasks
>
> We appreciate the reviewer’s enthusiasm about extending this approach to tasks like denoising, super-resolution, or event-based reconstruction. We agree that our spiking RetinexFormer framework is not limited to low-light enhancement. In principle, any continuous image-to-image mapping task could benefit from the neuromorphic efficiency if the architecture is adjusted appropriately. We envision applying similar SNN-based designs to tasks such as denoising (replacing the Retinex illumination estimation with, say, a noise estimation module, and then using a spiking U-Net for noise removal) or super-resolution (where the challenge would be to produce high-res outputs from low-res inputs – spikes could be used to gradually refine the image over time). The temporal coding approach could also be beneficial in event-based imaging or HDR reconstruction, where input arrives over time. While these extensions are outside the scope of the current paper, we will add a brief discussion in the conclusion or future work section noting that this is a first step toward neuromorphic low-level vision, and that the proposed Spike-RetinexFormer architecture could be adapted to other tasks.

---

> > ### Comment · Reviewer_da2C · 2025-08-04
> > **Official Comment by Reviewer da2C**
> >
> > The authors addressed my concerns and questions. I will raise my rating.

---

### Official Review · Reviewer_Phv2 · 2025-06-30

**Clarity:** 2
**Significance:** 2
**Originality:** 1
**Rating:** 3
**Confidence:** 4

**Summary:**

This paper introduces a new low-light image enhancement (LLIE) technique, named Spike-RetinexFormer, integrating spiking neural networks (SNNs) into a Retinex-based transformer framework (Retinexformer). By combining the energy-efficiency of SNNs and the enhancement performance of Retinexformer, the proposed method demonstrates its comparable or superior PSNR and SSIM performance compared to existing LLIE methods, while maintaining a compact model size and computational complexity.

**Questions:**

1. Please clearly clarify the unique contributions of Spike-RetinexFormer in Section 3, separating them from RetinexFormer baseline and SNN preliminaries.
2. Provide qualitative visual results on the SID, SMID, and SDSD datasets to ensure a comprehensive evaluation against existing LLIE methods.
3. Discuss the relative contribution of integrating SNN to Retinex-based LLIE framework in achieving performance improvements. Would user study or downstream task evaluation further validate the advantage of the proposed method versus Retinexformer baseline?

**Ethical Concerns:**

["NO or VERY MINOR ethics concerns only"]

**Final Justification:**

Based on the rebuttal and discussion, I slightly upgrade my score, but still on the fence. I am OK if this paper gets accepted but still think  the motivation of this work is not strongly supported.

**Limitations:**

Yes.

**Paper Formatting Concerns:**

None.

**Quality:**

2

**Strengths And Weaknesses:**

Strengths:
+ The proposed method Spike-RetinexFormer achieves improvements in PSNR and SSIM on most of the benchmark datasets compared to existing LLIE methods, especially outperforming the Retinexformer baseline.
+ The proposed method maintains a small model size and FLOPs.

Weaknesses:
- Section 3 intermixes preliminaries on Retinex-based LLIE framework and SNN techniques with the authors’ proposed contributions, making it difficult to follow.
- In Table 1, the usage of bold font is not explained. A footnote clarifying which metrics are highlighted, especially in the FLOPs and Params columns, is necessary.
- The RetinexFormer outputs shown in Figure 3 differ noticeably (e.g., color shifts) from those in the original paper [43]. The authors should clarify any differences in processing or rendering.
- Qualitative evaluations on the SID, SMID, and SDSD low-light datasets are absent, limiting the thoroughness of the comparative analysis.
- Many of the efficiency and quality gains appear attributable to the RetinexFormer backbone rather than the SNN integration, which diminishes the advantages of ANN-to-SNN transformation.
- A few typographical errors remain (e.g., L42 and L66), suggesting additional proofreading.

---

> ### Author Rebuttal · Authors · 2025-07-30
>
> ### Q1: Clarity of Section 3 (RetinexFormer vs. SNN vs. our contributions)
>
> We acknowledge that Section 3 in the original submission intermixes background and novel ideas. In the revised version, we will restructure Section 3 to clearly delineate the components. Section 3.1 will recap the RetinexFormer baseline and the one-stage Retinex-based framework (following the formulation from [43]), Section 3.2 will cover essential SNN preliminaries (LIF neurons, surrogate gradients, etc.), and Section 3.3 will detail our novel contributions in Spike-RetinexFormer. This includes the introduction of SpikingConvBlocks, the Spiking Illumination-Guided Attention (SIGA) module, and the integration of SNNs into the RetinexFormer architecture. By separating the RetinexFormer baseline and SNN background from our contributions, we will make it explicit which elements are prior work and which are introduced in Spike-RetinexFormer. This clarification will address the reviewer’s concern and make Section 3 easier to follow.
>
> ### Q2: Qualitative results on SID, SMID, and SDSD
>
> We apologize for this omission and will add qualitative results in a revised appendix, space permitting. We plan to include example enhanced images from Spike-RetinexFormer versus other methods on SID, SMID, and SDSD (indoor/outdoor) to complement the quantitative results (Table 2). These visual comparisons will highlight how our method handles the extreme noise and darkness in these datasets. As the paper has space constraints, we will provide these results in the supplementary material and clearly reference them to ensure a more comprehensive evaluation, as suggested.
>
> ### Q3: Contribution of SNN integration vs. RetinexFormer baseline
>
> We appreciate the reviewer’s concern that many gains might stem from the RetinexFormer backbone rather than the SNN component. Our results do indicate that the Retinex-based one-stage decomposition (the “ORF” from RetinexFormer) provides a strong foundation. However, the SNN integration *does* add value in terms of theoretical efficiency and noise suppression. In our ablation (Table 3), replacing standard attention with our spiking attention (SIGA) yielded an extra +0.21–0.43 dB PSNR improvement over non-spiking attention variants, confirming a performance benefit due to the spiking mechanism (because the spike-based attention effectively gates features under different illumination).
>
> More importantly, the SNN brings energy efficiency: many neurons do not fire, so a lot of computations are skipped, which would not be the case in an ANN. We will emphasize in the revision that on neuromorphic hardware the energy savings are significant, operations occur only on spike events rather than every pixel. As shown in table 4, using a shorter spike time window (T=4) reduced the simulated energy consumption to ~55% (ΔEnergy = 0.55×) of the baseline setting. These efficiency gains are a direct advantage of SNN integration.
>
> We agree that downstream task evaluation (feeding enhanced images into a detector, for example) could further demonstrate the practical benefit of our approach beyond PSNR/SSIM. Due to time constraints, we did not conduct a user study for this submission. However, we will note this as an avenue for future work. If the paper is accepted, we plan to include in a camera-ready version a brief discussion of how spike-based processing might better preserve perceptual quality (possibly referencing fewer flickering artifacts or more natural brightness adaptation, which could be validated by human viewers).
>
> We will also clarify that the primary contribution of Spike-RetinexFormer is the novel integration of SNNs into a RetinexFormer-like architecture, yielding comparable or better results while offering neuromorphic efficiency. In summary, we will strengthen the discussion of why SNNs add value (energy and potential robustness benefits) beyond the inherited RetinexFormer baseline.
>
> ### Q4: Formatting issues
>
> We will fix these issues in the revised manuscript. In Table 1, we had boldfaced certain figures (best metrics) without an explanation. We will add a footnote or caption clarification explaining the bold font (bold indicates the best result in each column). We will also thoroughly proofread the paper to correct the noted typographical errors (the errors at original Lines 42 and 66 and any others we find), ensuring a polished presentation is important. We appreciate these pointers, and these minor fixes will be made for the camera-ready version.
>
> ### Q5: Figure 3 output differences (RetinexFormer color shifts)
>
> The reviewer observed that the RetinexFormer results we showed in Fig. 3 have noticeable color differences compared to those in the original RetinexFormer paper. Our intention was to regenerate RetinexFormer’s outputs under our evaluation setting for a fair comparison. The differences likely arise from **implementation or processing details**: we used the official RetinexFormer code and applied the same processing across methods (without additional color correction). **It’s possible that the original paper’s figure applied certain post-processing or had a different pipeline for visualization, resulting in slightly different hues.** In the rebuttal and revised paper, we will clarify this point explicitly. We will state that we reran RetinexFormer on our test images and that minor color discrepancies versus the original paper’s published images may be due to differences in color space handling. To ensure transparency, we will double-check our RetinexFormer implementation against the official one. If we find any unintentional deviations (e.g., missing a specific calibration step), we will correct them in the final submission. Regardless, we will make sure the figure caption notes that our RetinexFormer results are obtained by us (and thus directly comparable to our method’s outputs). This clarification will resolve any confusion regarding Figure 3.

---

> > ### Comment · Reviewer_Phv2 · 2025-08-03
> >
> > Thanks the authors for the rebuttal. My concern on insufficient evaluation remains, which seems to be difficult to address via the limited time and space during rebuttal. And I agree with other reviewers that without clear analysis on energy advantage, the motivation of this work is not strongly supported.

---

> > > ### Author Response · Authors · 2025-08-07
> > > **Energy analysis**
> > >
> > > ***Q: Energy***
> > > ***A:***   We have specified the estimated energy consumption values for several models with different architectures and provided the estimation formula in the text below (You can also refer to our response to Reviewer vYvh).
> > >
> > > The energy consumption ($Energy$) of a network can be calculated using the following formulas:
> > >
> > > For ANNs, the theoretical energy consumption ($E_{ANN}$) is given by:
> > >
> > > $$E_{ANN} = E_{MAC} \times \text{FLOPs}$$
> > >
> > > where $E_{MAC}$ is the energy per Multiply-Accumulate (MAC) operation.
> > >
> > > For SNNs, the theoretical energy consumption ($E_{SNN}$) is given by:
> > >
> > > $$E_{SNN} = E_{AC} \times \text{SOPs}$$
> > >
> > > where $E_{AC}$ is the energy per spike-based Accumulate (AC) operation, and SOPs is the total number of Synaptic Operations. The number of Synaptic Operations for a given layer ($SOP^l$) can be calculated as:
> > >
> > > $$SOP^l = fr \times T \times \text{FLOPs}^l$$
> > > here, $fr$ is the firing rate of the block/layer, $T$ is the simulation time step for the spiking neurons, and $\text{FLOPs}^l$ refers to the floating point operations of the block/layer.
> > >
> > > In the final version of the paper, we will add a column for energy consumption values to Tables 2 and 3 in the experimental section to highlight the advantages of SNNs. We will correct other formatting issues in the final version of the paper. If there are any other issues that concern you, please point them out, and we will reply as soon as possible.
> > > We hope you can reconsider the score.

---

> > > ### Author Response · Authors · 2025-08-09
> > >
> > > Thank you for your time in reviewing our paper. Do you have any remaining concerns about our submission?

---

### Decision · Program_Chairs · 2025-09-17

**Decision:**

Accept (poster)

**Comment:**

The authors present a low light image enhancement algorithm, which integrated spiking neural networks in a Retinex-based transformer framework. The authors argue that this combination with spiking networks achieves comparable performance to existing methods.

Strengths identified by the reviewers include:

1. The authors demonstrate improvements in terms of peak signal to noise ratio (PSNR) and structural similarity (SSIM) metrics on most of the benchmark datasets.

2. The model is small, which can have implications for energy efficiency or edge computations

3. Some reviewers indicate that the writing is clear and the paper is well-organized.

4. Comparisons across a wide range of datasets are performed.

Weaknesses identified include:

1. Issues related to the organization of Section 3, which intermixes preliminaries on Retinex-based LLIE framework and SNN techniques with the authors’ proposed contributions.

2. Qualitative evaluations on the SID, SMID, and SDSD low-light datasets are absent

3. Uncertainty as to the extent to which the quality gains are attributable to the RetinexFormer backbone vs SNN integration.

4. Supporting better the energy-efficiency claims, especially with respect to actual throughput on hardware

The authors indicate that they will address these remaining points in the final version of their manuscript

The authors tried to address most of the reviewer concerns in their rebuttal. However not all concerns seem to have been addressed. In particular one reviewer still thinks there is insufficient evaluation and there is not a clear analysis of energy advantage. However the other reviewers' concerns with respect to inference latency and theoretical efficiency were mostly addressed which led these reviewers to raise their scores.

Overall the paper received 1 Borderline reject, 1 Borderline accept, 1 Accept and 1 Strong accept. Given that the remaining weaknesses will be addressed in the final manuscript, I am leaning towards recommending acceptance of the paper.